# Adverse Childhood Experiences and Mental Health among Students Seeking Psychological Counseling Services

**DOI:** 10.3390/ijerph20105906

**Published:** 2023-05-22

**Authors:** Francesco Craig, Rocco Servidio, Maria Luigia Calomino, Francesca Candreva, Lucia Nardi, Adriana Palermo, Alberto Polito, Maria Francesca Spina, Flaviana Tenuta, Angela Costabile

**Affiliations:** 1Department of Cultures, Education and Society (DICES), University of Calabria, 87036 Cosenza, Italy; 2Psychological Counseling Services, University of Calabria, 87036 Cosenza, Italyadrianapalermo23@gmail.com (A.P.);

**Keywords:** psychological counseling, university students, psychological distress, coping, personality

## Abstract

Recent years have seen a marked rise in the number of students accessing University Psychological Counseling (UPC) services, and their concerns have been increasingly severe. This study aimed to examine the impact of cumulative adverse childhood experiences (ACEs) on mental health in students who had approached counseling services (N = 121) and students who had no experience with counseling services (N = 255). Participants completed an anonymous online self-report questionnaire measuring exposure to adverse childhood experiences (ACE-Q), psychological distress (General Anxiety Disorder-7 (GAD-7) and Patient Health Questionnaire-9 (PHQ-9), personality traits (PID-5), and coping strategies. We found that students who approached UPC services scored higher on cumulative ACEs than the non-counseling group. While ACE-Q score was a direct positive predictor of PHQ-9 (*p* < 0.001), it did not predict GAD-7. Moreover, the results supported the existence of a mediation effect of avoidance coping, detachment, and psychoticism on the indirect effects of ACE-Q score on PHQ-9 or GAD-7. These results underlined the importance of screening for ACEs in a UPC setting because it can help identify students at higher risk for developing mental and physical health problems and provide them with early interventions and support.

## 1. Introduction

A significant body of literature supports trauma-informed approaches to mental health care on the widespread occurrence of adverse childhood experiences (ACEs) and their negative effects on mental health and overall well-being as a significant public health concern [1]. ACEs refer to a range of traumatic experiences that occur in childhood, such as physical, emotional, or sexual abuse; neglect; and exposure to household dysfunction, such as substance abuse, domestic violence, or mental illness [2,3]. Results from several studies suggest that individuals with a history of ACEs are at an increased risk for developing a range of mental health problems in adulthood, such as depression, anxiety, post-traumatic stress disorder (PTSD), substance abuse disorders, as well as a higher risk of engaging in self-harm behaviors and suicide [3,4,5,6].

Previous studies have proved that individuals who seek mental and psychological health services have a higher occurrence of past traumatic experiences [7,8]. In recent years, there has been a steady increase in the number of students seeking University Psychological Counseling (UPC) services, and the concerns they bring to these services have become increasingly severe [9,10]. Many university students might have experienced ACEs in their childhood, and these negative experiences could affect their ability to succeed academically and socially. However, the effects of ACEs on mental health in university students have not been fully explored. A recent study found that childhood adversities negatively impacted mental and physical health and quality of life in a sample of 858 university students [11]. Watt and colleagues [12] detected that university students who reported a history of ACEs were more likely to show symptoms of depression and anxiety than those without a history of ACEs. Interestingly, they found that neuronal health and inflammation conditions were associated with mental health disorders for individuals who had experienced four or more ACEs, but this was not the case for those without these childhood experiences.

Moreover, other previous studies also highlighted that ACEs were associated with lower academic success and increased the risk of dropping out of university [13,14]. University students who have experienced ACEs are also more likely to engage in risky behaviors, such as substance abuse and unprotected sex, compared to their peers without a history of ACEs [11,15]. Therefore, proactively identifying and managing ACEs could help reduce their negative impact on university students’ psychological distress. Indeed, UPC services could be crucial in supporting students’ well-being. University counseling services can provide professional support and resources to students to manage and cope with their ongoing ACEs-related challenges. Thus, the services offered by a university can help students develop healthy relationships with peers, improve self-esteem, overcome personal obstacles, build resilience, reduce psychological distress, and prevent more severe issues from arising [16,17].

However, the relationship between ACEs and psychological well-being is complex and multidimensional, involving several factors that can contribute to the risk of developing symptoms of psychopathology. Therefore, studying other variables that may be involved in this relationship is crucial to gain a more comprehensive understanding of the impact of ACEs on students’ mental health.

Research indicates that exposure to ACEs can increase the risk of developing personality disorders, such as borderline personality disorder and narcissistic personality disorder [18,19]. This is likely because ACEs can lead to the formation of maladaptive personality traits, such as impulsivity, emotional dysregulation, and negative self-concept. ACEs can also have a negative impact on the development of personality traits, such as emotional stability, extraversion, agreeableness, and conscientiousness [20]. For example, individuals who have experienced ACEs may be more likely to score lower in emotional stability and may be more prone to negative emotions, such as anxiety and depression. In addition, some studies showed that individuals who have experienced trauma may have increased levels of emotional dysregulation, self-esteem difficulties, and impulse control difficulties and may be more likely to develop avoidant attachment styles [21,22]. It is important to note that the effects of trauma on personality traits can vary depending on the type of trauma experienced, the severity of the trauma, and an individual’s coping mechanisms and resilience. Grusnick and colleagues [23] detected that the higher the ACE score, the greater the effects on personality, emotions, and affect factors, and certain types of ACEs, particularly abuse, have a stronger impact. This supports the idea that the cumulative risk of multiple ACEs leads to more significant adverse outcomes.

Exposure to ACEs might also lead to a chronic stress response, which could affect the functioning of the hypothalamic–pituitary–adrenal (HPA) axis and the development of the brain [24,25]. These brain structure and function changes could affect an individual’s ability to cope with stress and regulate emotions. Some coping strategies that are commonly studied with ACEs are problem-focused (PF) coping and avoidant emotion-focused (AEF) coping [26]. PF coping focuses on finding solutions to problems and building self-confidence; AEF coping involves strategies that aim to reduce negative emotions in response to stressors but do not address the underlying problem [26,27]. Research has shown that exposure to ACEs is associated with less frequent use of PF coping and greater use of AEF coping [28,29]. Further, studies showed that adolescents who have experienced ACEs are more likely to use avoidance coping strategies than those who have not experienced ACEs [30]. However, in the long term, relying on avoidance to cope with stressors can lead to psychopathological symptoms [31,32]. Thus, studying coping strategies in university students is important because the transition to university life can be challenging for many of them. In addition, understanding how university students, especially those with a history of negative experiences, cope with challenges, such as academic stress, can help identify those at risk for mental health problems and provide them with the support they need to succeed.

Considering all available evidence seems to show that cumulative ACE exposure may negatively influence personality functioning and coping strategies, thereby leading to even more significant psychological distress and vulnerability. However, the relationship between ACEs and mental health is multidimensional and intricate. Therefore, a better understanding of this relationship could aid UPC services’ design and development of more effective prevention strategies, intervention programs, and appropriate support for students who want to resolve any issues related to their university experience. Thus, we aimed to assess the prevalence of cumulative ACE exposure and psychological distress among a sample of Italian university students. Specifically, we hypothesized that (a) the prevalence of ACEs would be higher among students who had counseling sessions compared to students who did not have experience with counseling sessions at and outside of the university; (b) cumulative ACE exposure would be related to psychological distress symptoms (anxiety and depression); and (c) maladaptive personality traits and coping strategies might mediate the relationship between cumulative ACE exposure and psychological distress symptoms.

## 2. Materials and Methods

### 2.1. Study Design

A cross-sectional, web-based survey was carried out among undergraduate students at the University of Calabria (Italy) between September and December 2022. Convenient response sampling was used to collect data through questionnaire links. The participants were invited to participate in the online survey by filling out the Google Forms. The students received the survey’s link through a QR code, email, and advertisements through the university’s psychological counseling service, courses, and on-campus advertisements. The participants did not receive any reward. The data collected through the online questionnaire were completely anonymous, and responses were used for statistical purposes only, in compliance with the European Data Protection Regulation GDPR 679/2016 to protect privacy. The Ethics Committee of the University of Calabria approved the study (Protocol number: 0066896—September 2022), and we obtained written informed consent from all subjects enrolled. This study was conducted in accordance with the Declaration of Helsinki.

### 2.2. Participants

The counseling group (N = 121) consisted of university students who had counseling sessions at our UPC service. To book a counseling session, students had to reserve through an online ticketing service. We recruited students who enrolled in Humanities, Technology, Medicine, Pharmacy, Social Services, and Education Sciences degree courses. Students who sought counseling were invited to participate in this study immediately after the first clinical interview. It was conveyed to the participants that accepting or declining participation in this study would not affect the organization of their psychological support session cycle. The estimated time to complete the questionnaire was approximately 15–20 min.

The inclusion criteria were being 18 years or older, having the ability to comprehend and read a self-report questionnaire, and having sought psychological support exclusively at the university counseling service. As the exclusion criteria, we did not include students with a significant mental (e.g., schizophrenia, bipolar disorder, psychosis, or history of substance abuse) or physical (e.g., neurological disorder or chronic medical conditions) condition affecting their ability to complete the questionnaire accurately. The non-counseling group included students (N = 255) who did not have experience with counseling sessions at and outside of the university. The QR code was distributed to the students included in the non-counseling group in their respective classrooms after an orientation session for filling the questionnaire by one of the investigators.

### 2.3. Measures

The online survey with 118 questions consisted of sections capturing information on (1) informed consent and data handling; (2) socio-demographic indicators, such as age, gender, relationship status (single, engaged, married, separated, or widowed), residential status (university residence, living with family, apartment, apartment with roommates, or other arrangements), type of degree program, and year of university enrollment; (3) exposure to adverse childhood experiences (ACE Questionnaire—ACE-Q); (4) anxiety (General Anxiety Disorder-7—GAD-7) and depression symptoms (Patient Health Questionnaire-9—PHQ-9); (5) personality traits (Personality Inventory for DSM Brief Version—PID-5-BF); and (6) coping strategies (Brief-COPE).

Cumulative ACE exposure was measured using the ACE-Q [33]. The ACE-Q is widely used in public health and medical research, as well as in clinical settings, to help identify and address the impact of childhood trauma on a person’s health and well-being. The questionnaire, created by Dr. Vincent Felitti and his team in 1998, is a 10-question survey that evaluates childhood trauma. The questions assess ten experiences children might have faced, including physical, emotional, or sexual abuse; neglect; household dysfunction; and other stressful life events. The participants were asked to respond to each item on the questionnaire with either a “yes” or “no”. A positive response to any question indicated that the participants had experienced at least one type of ACE. The ACE total score was calculated as a continuous variable by assigning one point to each affirmative answer and then summing up the points. The score could range from 0 to 10, with higher scores indicating exposure to more ACEs. A minority (5–10%) of the general population score 4 or more, where the general long-term health consequences become most pronounced [34]. Validation research indicates that the ACE-Q score remains a reliable predictor despite potential biases from historical self-reported memory biases [35]. The Cronbach’s α reliability coefficient for the entire ACE-Q scale was 0.73, demonstrating relatively high reliability [36].

The presence of symptoms of generalized anxiety was evaluated using the GAD-7 [37]. It is a 7-item self-report questionnaire designed to assess symptoms and screen for generalized anxiety disorder. The items on the scale ask individuals to rate the frequency and severity of symptoms they have experienced in the past two weeks, such as excessive worry, trouble relaxing, and irritability. Scores on the GAD-7 range from 0 to 21, with higher scores indicating greater severity of GAD symptoms. The following cut-off scores are commonly used to categorize individuals into different levels of anxiety: a score between 0 and 4 (minimal anxiety); between 5 and 9 (mild anxiety); between 10 and 14 (moderate anxiety); and between 15 and 21 (severe anxiety). These cut-off scores are based on research and clinical experiences and are widely used in clinical and research settings. The GAD-7 is a reliable and valid tool for assessing GAD symptoms and is widely used in both clinical and research settings [37]. We found that the GAD-7 had good reliability, with Cronbach’s α = 0.84.

The presence of depression symptoms was evaluated using the PHQ-9 [38]. It consists of 9 questions that assess symptoms of depression, such as loss of interest, changes in appetite, and feelings of hopelessness. The answers to each question are rated on a 4-point scale, with higher scores indicating more severe depression. The cut-off scores indicate the threshold for depression, with scores of 10 and above suggesting a significant level of depression. The levels of depression according to the PHQ-9 are mild (5–9), moderate (10–14), moderately severe (15–19), and severe (20–27). The PHQ-9 is a commonly used tool in clinical settings and is considered a reliable and valid measure to assess levels of depression severity [38]. In the current study, the value of the reliability coefficient Cronbach’s α for the overall PHQ-9 scale was 0.83, indicating excellent reliability.

Personality traits were evaluated using the Brief Form of PID-5 [39]. It is a 25-item measuring tool evaluating an individual’s personality traits and disorders in accordance with the DSM-5 (*Diagnostic and Statistical Manual of Mental Disorders*) criteria. The test measures five broad personality domains: negative affectivity, detachment, antagonism, disinhibition, and psychoticism. Each domain consists of several facets that further describe individual personality traits. It is a self-report questionnaire which is answered on a five-point Likert scale ranging from “Not at all true of me” to “Very true of me”. Each trait domain is composed of 5 items, and the scale is self-administered, allowing a subject to assess how accurately each statement describes them in general. We found that the PID-BF had good reliability (Cronbach’s α was 0.83).

The Brief-COPE [40,41] was used to assess a broad range of coping responses, and it was developed as a short version of the original 60-item COPE scale. The Brief-COPE is a 28-item self-report questionnaire designed to measure effective and ineffective ways to cope with stressful life events. This scale is a helpful tool in counseling settings to identify an individual’s coping styles in response to stressors. The scale can determine someone’s primary coping styles with subscales that measure problem-focused strategies (active coping, planning, and use of instrumental support), emotion-focused strategies (use of emotional support, positive reframing, acceptance, religion, and humor), and avoidance coping (self-distraction, denial, substance use, behavioral disengagement, venting, and self-blame). The items are rated on a 4-point Likert scale, ranging from “not at all” to “a lot”. We used the total scores of each subscale, calculated by summing the appropriate items for each subscale, to level out the scores from the various tests. We found a satisfactory internal consistency, with Cronbach’s alpha of 0.77.

### 2.4. Data Analysis

We did not have missing data as the online questionnaire required the participants to input responses for missed items. The sample size for the current study was established using a priori statistical power analysis with the support of the G-Power 3.1 software. Assuming a mean effect size f2 of 0.15, with a statistical power level of 0.95, alpha of 0.05, and a maximum of 12 predictors, we calculated a minimum required sample size of 184. First, we inspected the reliability of the administered instruments using Cronbach’s alpha (α) reliability coefficient. Then, descriptive statistics and Pearson’s r correlations were computed to explore the variables’ properties and determine their relationships. The scale scores were normally distributed, as the most significant value for skewness was 1.31 (for the ACE non-counseling group), and the most significant value for kurtosis was 2.24 (for the ACE non-counseling group), which could be considered below the thresholds for non-normality [42]. The Chi-squared (χ2) test was used to compare differences among categorical variables, including gender, relationship status, residential status, frequency of ACE type, and levels of GAD-7 and PHQ-9. The independent *t*-test was performed to explore the differences between the counseling and non-counseling groups. To assess the hypothesized relationships between variables, we tested a path model with observed variables, where ACE-Q was the predictor; coping strategies and personality traits (PID-5-BF) were the mediators; and psychological distresses (GAD-7 and PHQ-9) were the outcome. Direct paths from the predictor to the outcome variables were estimated, as well as correlations between the control variables, the mediators, and the outcome variables.

Moreover, a bootstrapping procedure using 5000 resamples was performed to test the significance of indirect effects with a 95% confidence interval (CI) [43]. In estimating all path coefficients, we controlled for group, age, and gender. However, to improve the readability of the path model, we reported only some results of the control variables.

Since we tested a saturated model (path analysis), which is expected to obtain a perfect fit, a trimmed model was examined (e.g., the model in which non-significant paths were constrained to zero). Therefore, in establishing model fit, we used the comparative fit (CFI), Tucker–Lewis (TLI), and root-mean-square error of approximation (RMSEA) indexes. According to Kline, we considered values of CFI ≥ 0.95, TLI ≥ 0.95, and RMSEA ≤ 0.05 as indicators of good model fit. Finally, we computed each mediation effect size (PM) as the ratio of the indirect effect to the total effect [44].

The statistically significant *p*-value was set at 0.05. All statistical analyses were carried out with the support of the SPSS (version 28) and R-lavaan (version 0.6-13) packages.

## 3. Results

### 3.1. Descriptive Statistics

The online survey was administered to 376 undergraduates, of which 121 were included in the counseling group and 255 in the non-counseling group. The sample characteristics are summarized in Table 1.

The mean age of the counseling group was 21.96 ± 2.9 years, ranging from 18 to 35 years; the mean age of the non-counseling group was 21.41 ± 3.9 years, ranging from 18 to 50 years. No significant difference was found between the groups regarding age (*p* = 0.385). Comparing the socio-demographic characteristics of the two groups, we identified a statistically significant gender difference (*p* = 0.002). We found more males in the counseling group than in the non-counseling group. There were no significant differences between the groups in terms of relationship status and residential status. The observed and expected counts are presented together in Table 1.

### 3.2. ACEs

There were significant differences between the groups in the mean score (*p* < 0.001) of the ACE-Q and the prevalence of ACE types (see Table 2). Overall, 35% of students in the counseling group scored four or above on the ACE-Q, compared to 20% in the non-counseling group. The average ACE-Q score in the counseling group was 2.26 with a standard deviation of 1.74, while the average score of the non-counseling group was 1.57 with a standard deviation of 1.67 (see Table 2). We observed that in the counseling group, 54% of students were exposed to emotional abuse, 24% to physical abuse, 10% to sexual abuse, 60% to emotional neglect, 9% to physical neglect, 13% to parental divorce, 13% to battered mother, 2.5% to growing up with substance abuse, 31% to growing up with substance abuse, 31% to parental mental illness, and 7% to imprisoned parents. In the non-counseling group, we found that 39% of students were exposed to emotional abuse, 15% to physical abuse, 16% to sexual abuse, 31% to emotional neglect, 7% to physical neglect, 10% to parental divorce, 7.5% to battered mother, 8% to growing up with substance abuse, 18% to parental mental illness, and 5% to imprisoned parents. Furthermore, we observed (see Figure 1) that a larger proportion of students in the counseling group were more exposed to adverse experiences, such as emotional abuse (*p* = 0.005), physical abuse (*p* = 0.041), emotional neglect (*p* < 0.001), growing up with substance abuse (*p* = 0.033), and mental illness (*p* = 0.003).

### 3.3. Psychological Distress Symptoms

Regarding psychological distress levels, the prevalence of anxiety and depressive levels and the mean scores of GAD-7 and PHQ-9 are reported in Table 1 and Table 2. Our findings indicate that the counseling group had a higher proportion of students with severe levels of anxiety or depression compared to the non-counseling group, with 35% of the counseling group and 20% of the non-counseling group reporting severe anxiety, and 9% of the counseling group and 1% of the non-counseling group reporting severe depression symptoms. In addition, we analyzed the differences between the two groups in terms of GAD-7 and PHQ-9 mean scores. We detected a significant difference between the two groups in terms of psychological distress, with the counseling group displaying higher symptoms of anxiety (*p* < 0.001) and depression (*p* < 0.001) compared to the non-counseling group.

### 3.4. Personality Traits and Coping Strategies

This section presents the statistical and comparative analyses of personality facets and coping strategies. Several personality traits were shown as dysfunctional, and the group variable was used to control the final comparison to present individual performance. It was found that students seeking psychological counseling service scored significantly higher than students in the non-counseling group on negative affect (*p* = 0.004), detachment (*p* < 0.001), antagonism (*p* = 0.014), and psychoticism (*p* = 0.003). Comparing scores on various scales of coping strategies, we found that the counseling group had higher scores on the emotion-focused coping (*p* < 0.001) and avoidance coping (*p* < 0.001) scales compared to the non-counseling group (see Table 2).

### 3.5. Correlation and Mediation Analyses

The results of the correlation analysis (see Table 3) suggest that cumulative ACE exposure is significantly correlated with generalized anxiety disorder (*p* = 0.003), symptoms of depression (*p* < 0.001), each personality trait, and coping strategies (except problem-focused coping).

The bootstrapped regression-based path analysis displayed the direct and indirect effects. The results of the estimated direct effects for the saturated model are shown in Figure 2, whereas the results of the estimated indirect effects are presented in Table 4. ACE-Q score is a direct positive predictor of PHQ-9 score (*p* < 0.001) but not for GAD-7 score (*p* = 0.909). As regards the indirect effects of ACE-Q score on GAD-7 score, the current results supported the existence of a full mediation effect passing through avoidance coping (*p* = 0.019). In the second indirect association from ACE-Q score to PHQ-9 score, we found a partial mediation effect passing through avoidance coping (*p* = 0.027), detachment (*p* = 0.011), and psychoticism (*p* = 0.005) (see Table 4). The effect sizes of the indirect effects (PM) are satisfying.

Concerning the control variables, the counseling group negatively affected the ACE-Q score (B = −0.74, 95% CI [−0.30, −0.10], β = −0.20). Moreover, the covariate group was negatively associated with all other variables of the proposed model, although these paths are not represented in Figure 2 to improve readability. Gender (being female) showed a positive effect on negative affectivity (B = 1.83, 95% CI [0.18, 0.37], β = 0.28) and ACE−Q score (B = 0.46, 95% CI [0.02, 0.19], β = 0.11); conversely, we found a negative association between gender (being male) and antagonism (B = 0.65, 95% CI [−0.21, −0.04], β = −0.11). Furthermore, age showed a negative effect on negative affectivity (B = −0.11, 95% CI [−0.23, −0.07], β = −0.15), avoidance coping (B = −0.10, 95% CI [−0.20, −0.02], β = −0.12), and psychoticism (B = −0.15, 95% CI [−0.27, −0.10], β = −0.19). No remaining significant effects emerged.

Finally, we explored the model’s fit after removing the non-significant paths for the main effects. Specifically, we trimmed the tested model by fixing non-significant paths to zero (e.g., dashed lines in Figure 2). As a result, the revised model showed excellent fit (χ^2^ (8) = 12.123, *p* = 0.145, CFI = 0.997, TLI = 0.968, RMSEA = 0.037 (90% CI: 0.00, 0.008), SRMR = 0.018).

## 4. Discussion

The present study examined cumulative ACE exposure’s role on university students’ psychological distress. As hypothesized, we found higher rates of cumulative ACE exposure in students who sought counseling services than those who did not. Specifically, the students in the counseling group were more likely to have undergone adverse experiences, including emotional abuse, physical abuse, emotional neglect, growing up in a household affected by substance abuse, and exposure to mental illness. These results further support the idea that individuals seeking mental health and psychological support, such as UPC services, have a higher rate of previous adverse experiences, both in their childhood and adulthood, than the general population [45,46]. This highlights the importance of addressing and treating the effects of adverse experiences in the university context to improve the well-being and academic performance of students seeking counseling services.

Another important finding is that cumulative ACE exposure is significantly related to psychological distress, which is consistent with previous research exploring the relationship between ACEs and outcomes in adulthood [47]. We hypothesized that cumulative ACE exposure could be a predictor of psychological distress, but this hypothesis was not fully confirmed. The results indicated that ACE-Q score was a direct positive predictor of PHQ-9 score but not GAD-7 score. This finding supports the idea of a linear relationship between cumulative ACE exposure and the likelihood of developing depression later in life [48,49,50]. There are several possible explanations for this link. A previous study reported that child maltreatment is associated with comorbid conditions of depression and inflammation (C-reactive protein—CRP) in adults with a history of child maltreatment, but not among those without such experiences [51]. Similarly, a study suggested that neuroendocrine changes secondary to early life stress likely reflect the risk of developing depression in response to stress, potentially due to the failure of a connected neural circuitry implicated in emotional, neuroendocrine, and autonomic control to compensate in response to challenges [52]. More recently, Watt and colleagues [12] found that students who had experienced four or more ACEs showed a correlation between inflammation (CRP) and neuronal health (brain-derived neurotrophic factor—BDNF) with mental health disorders, but this was not the case for students without a history of ACEs [12]. Furthermore, research has proposed that epigenetic mechanisms, such as histone acetylation and DNA methylation, could play a role in the lasting increase in depression risk after experiencing adverse life events, thus providing a framework to integrate genetic and environmental factors [53,54,55]. A model emerges from these studies suggesting that genetic and environmental risk factors and their interactions could trigger abnormal epigenetic mechanisms that target stress response pathways, neuronal plasticity, and other behaviorally relevant pathways linked to major depression [56]. Thus, repeated adversity during critical developmental stages can modify brain chemistry and structure, as well as endocrine, immune, and metabolic systems, leading to an increased risk of mental health disorders, such as mood disorders [52,57,58].

Consistent with our third hypothesis, personality traits and/or coping strategies are another mechanism through which ACEs may affect later life mental health outcomes. Specifically, detachment and psychoticism traits were found to partially mediate the link between cumulative child maltreatment and symptoms of depression. Although personality functioning is primarily examined in the context of personality disorders, our findings highlight its significance for mental health outcomes following childhood maltreatment or abuse. These results support the view expressed by Krakau and colleagues [20] that impaired personality functions should not be disregarded in individuals who have experienced trauma. Moreover, we detected a distinct association between cumulative ACE exposure with detachment and psychoticism domains and avoidance coping strategies. The data in the literature have shown that these personality domains comprise trauma-specific (maladaptive) coping strategies, such as avoidance, which exhibit considerable conceptual and empirical similarities with PTSD or mood disorders. Hence, as suggested by Back and colleagues [59], we deduce that childhood trauma could imprint on personality traits, which eventually manifest as long-term, trauma-reactive internalizing coping strategies.

Interestingly, our finding revealed that the indirect effect of ACE-Q score on GAD-7 score was fully mediated by avoidance coping. This suggests that students who have experienced higher levels of ACEs may be more likely to develop anxiety symptoms due to their use of avoidance coping strategies. These results are consistent with previous research showing that avoidance coping strategies can exacerbate anxiety symptoms [27,60]. Avoidance is commonly viewed as an ineffective behavioral response to excessive fear and anxiety, ultimately perpetuating the presence of anxiety disorders. The *Diagnostic and Statistical Manual of Mental Disorders-5* (DSM-5) suggests the implicit assumption that avoidance behaviors represent key features of all anxiety disorders, with fear and anxiety linked to the disorder closely associated with avoidance tendencies [61]. However, avoidance coping is often, but not always, a maladaptive coping response. In the short term, denial, for example, may be beneficial early on in a traumatic episode or if it occurs in a situation that is both uncontrollable and too threatening. It is not likely beneficial over the long term because it does not effectively target the threat and its impact. For example, ongoing avoidance of a traumatic experience is not an adaptive form of emotion regulation in the long term. In contrast, distancing, or efforts to disengage from a situation temporarily in an attempt to diminish its significance, can be an adaptive emotion-focused disengagement strategy [62]. In this context, our data highlight that students with adverse experiences may engage in avoidance coping strategies, which, in the long term, can lead to the emergence of anxiety symptoms that are the primary reason for seeking counseling services. Thus, interventions aimed at reducing avoidance coping and promoting more adaptive coping strategies, such as problem solving or emotion regulation, may be particularly beneficial for students who have experienced childhood adversity during psychological and behavioral interventions provided by UPC services.

### 4.1. Implications for University Psychological Counseling and Future Directions

ACEs have significant long-term associations with psychosocial outcomes of university students, including poorer mental health [11]. The role of cumulative ACE exposure on the psychological distress of university students was examined. The current findings suggest that individuals seeking mental health and psychological services have a higher rate of previous adverse experiences than students who did not have experience with counseling sessions. These results highlight the importance of addressing and treating the effects of adverse experiences in the university context to improve students’ well-being and academic performance. Hence, UPC services should play a crucial role in identifying and addressing the effects of ACEs on students’ mental health. Screening for ACEs in a UPC setting is crucial because it can help identify students at higher risk for developing mental and physical health problems and provide them with early interventions and support. Screening can also help raise awareness about the prevalence of ACEs and the importance of addressing their impact on students’ overall health and well-being.

Further research should be undertaken to find ways to screen all university students, not just those who seek out services. We should not dismiss the possibility that some students who experienced early adverse experiences may feel a sense of self-blame or shame, thus preventing them from seeking professional help for their psychological and emotional well-being [63]. Individuals who do not receive appropriate support and treatment for their mental health concerns may struggle to cope with the academic demands of university life, leading to poor academic performance, lower engagement levels, and higher dropout rates. Thus, it is crucial to identify and address existing external barriers (organizational and institutional) to make UPC services available for the student population to ensure that students have access to psychological support.

Furthermore, the current results highlight how UPC services can help university management embrace a broader culture of wellness and rethink their approach to student support. Universities are facing a surge in demand for assistance that far exceeds their capacity, and it becomes increasingly clear that the traditional counseling center model is ill-equipped to solve the problem. Some of the reasons for this increase are positive. Compared to previous generations, more students today have access to mental health support services before college, suggesting that higher education is now an option for a broader segment of society. The stigma around mental health issues also continues to decrease, enabling more people to seek help rather than suffer in silence. However, students must also navigate a dizzying array of challenges, ranging from coursework, relationships, and adjustment to campus life to economic strain, social injustice, mass violence, and various forms of loss related to COVID-19 [64,65,66]. As a result, universities nationwide are adopting approaches such as group therapy, peer counseling, and telemedicine. In the context of universities, a stepped-care model could be particularly effective for addressing students’ diverse mental health needs [67]. This model should include different levels of care, ranging from self-help resources to individual counseling or therapy, based on the severity of a student’s condition. For students who are struggling with more severe mental health issues, such as depression or anxiety disorders, individual counseling or therapy sessions can be incredibly beneficial. This level of care provides a safe and confidential space for students to explore their thoughts and feelings with trained specialists and learn coping strategies for managing their symptoms. However, not all students may require this level of care. For many students, a less intensive approach may be sufficient to address their mental health needs. This is where peer counseling and workshops on stress, sleep, time management, and goal setting can be extremely valuable. Creating a safe and supportive campus environment can also help students cope with the effects of ACEs and promote resilience. Universities are responsible for supporting their students’ mental health and well-being. Thus, UPC services should establish an evidence-based protocol to promote a culture of health and well-being among university students through implementing good prevention and care practices.

### 4.2. Limitations

Some limitations deserve mention. First, this study’s sample was limited to students at a single university, which implies that even with many participants, it may not be a representative or generalizable sample of all college students or all adults in the country. Second, this study did not investigate the distinct impacts of different ACE profiles. The counseling and non-counseling groups exhibited a high degree of heterogeneity in terms of the specific types of adverse experiences they had encountered, which rendered it impossible to conduct subgroup comparisons. Third, the use of the ACE-Q might introduce potential memory errors because the questionnaire relies on individuals’ recall of past events, which may be subject to memory biases and inaccuracies. Moreover, the ACE-Q does not assess the severity, frequency, or duration of adverse experiences, and only captures their dichotomous nature, which may limit its ability to fully capture the impact of these experiences on individuals. We also did not assess other factors, such as the social support of university students, that might impact their psychological well-being. For example, several studies showed that perceived social support could moderate the relationships between adverse effects of stressful events and mental health outcomes in clinical and non-clinical populations [68,69]. Finally, we did not assess resilience skills. Research has revealed psychological resilience to be an effective mediator of the relationship between ACEs and negative mental health outcomes as people with low levels of resilience may have more difficulty in recovering from the effects of ACEs [70].

## 5. Conclusions

Overall, these findings contribute to our understanding of the complex interplay between adverse childhood experiences, coping strategies, personality domains, and psychological outcomes. The current results reveal that ACEs have a negative association with students’ psychological distress, and personality traits and/or coping strategies are mediators through which ACEs may affect later life mental health outcomes. Hence, acknowledging the association between ACEs and mental health in university students and providing appropriate support is crucial for promoting the well-being and success of all students. Early identification and intervention by UCP services could significantly reduce the negative effects of ACEs on the mental health and well-being of students. Therefore, raising awareness of the importance of counseling services and encouraging their use among university students is essential. By doing so, universities can better support their students’ mental health and well-being and help them achieve their academic and personal goals.

## Figures and Tables

**Figure 1 ijerph-20-05906-f001:**
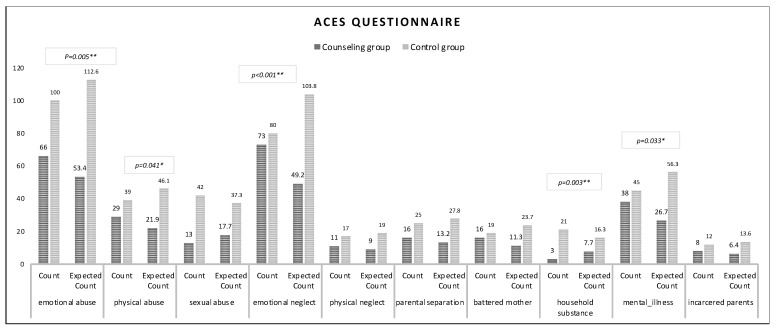
Observed and expected frequencies of each ACE type. Note: * *p*-value < 0.05 and ** *p*-value < 0.01.

**Figure 2 ijerph-20-05906-f002:**
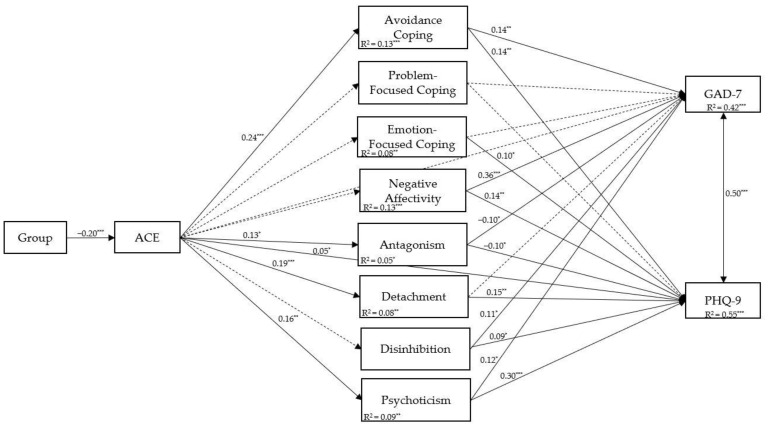
Results of the path model with standardized effects. Note. * *p*-value < 0.05, ** *p*-value < 0.01, and *** *p*-value < 0.001.

**Table 1 ijerph-20-05906-t001:** Prevalence and differences among categorical variables between groups (observed vs expected value).

Variable		Counseling Group (N = 121)	Non-Counseling Group (N = 255)	Value	*p*-Value
Gender				9.841	0.002 **
Male	Count	37	42		
	Expected count	25.4	53.6		
	% within groups	30.6%	16.5%		
Female	Count	84	213		
	Expected count	95.6	201.4		
	% within groups	69.4%	83.5%		
Relationship status				1.568	0.667
Single	Count	61	121		
	Expected count	59	123		
	% within groups	52.1%	49.6%		
Engaged	Count	56	120		
	Expected count	57	119		
	% within groups	47.9%	49.2%		
Married	Count	0	1		
	Expected count	0.3	0.7		
	% within groups	0.00%	0.4%		
Separated	Count	0	2		
	Expected count	0.6	1.4		
	% within groups	0.00%	0.8%		
Residential status				2.686	0.443
University residence	Count	16	28		
	Expected count	13.9	30.1		
	% within groups	14.8%	12.00%		
Apartment	Count	5	6		
	Expected count	3.5	7.5		
	% within groups	4.6%	2.6%		
Living with family	Count	56	115		
	Expected count	54.2	116.8		
	% within groups	51.9%	49.4%		
Apartment with roommates	Count	31	84		
	Expected count	36.4	78.6		
	% within groups	28.7%	36.1%		
GAD-7 level				22.569	<0.001 **
Minimal	Count	3	28		
	Expected count	10	21		
	% within groups	2.5%	11.00%		
Mild	Count	30	103		
	Expected count	42.8	90.2		
	% within groups	24.8%	40.4%		
Moderate	Count	46	74		
	Expected count	38.6	81.4		
	% within groups	38.00%	29.00%		
Severe	Count	42	50		
	Expected count	29.6	62.4		
	% within groups	34.7%	19.6%		
PHQ-9 level				51.259	<0.001 **
N	Count	4	51		
	Expected count	17.7	37.3		
	% within groups	3.3%	20.00%		
Mild	Count	35	117		
	Expected count	48.9	103.1		
	% within groups	28.9%	45.9%		
Moderate	Count	45	63		
	Expected count	34.8	73.2		
	% within groups	37.2%	24.7%		
Moderately severe	Count	26	21		
	Expected count	15.1	31.9		
	% within groups	21.5%	8.2%		
Severe	Count	11	3		
	Expected count	4.5	9.5		
	% within groups	9.1%	1.2%		

Note: YES (Y) and NO (N); General Anxiety Disorder-7 (GAD-7), Patient Health Questionnaire-9 (PHQ-9); ** *p*-value < 0.01.

**Table 2 ijerph-20-05906-t002:** Mean scores, standard deviations, skewness, Kurtosis, and differences in clinical variables between groups.

	Mean (SD)		Skewness		Kurtosis				95% CI	
	Counseling	Non-Counseling	Counseling	Non-Counseling	Counseling	Non-Counseling	t	*p*-Value	Lower	Upper
ACE-Q	2.26 (1.74)	1.57 (1.67)	0.39	1.31	−0.84	2.24	3.676	<0.001 **	0.32	1.055
GAD-7	12.46 (4.37)	9.97 (4.49)	−0.13	0.29	−0.67	−0.82	5.077	<0.001 **	1.528	3.46
PHQ-9	12.58 (5.24)	8.22 (4.30)	0.49	0.64	−0.30	0.29	8.541	<0.001 **	3.355	5.362
PID-5										
Negative affectivity	9.12 (2.55)	8.27 (2.68)	−0.12	−0.23	−0.39	−0.19	2.915	0.004 **	0.276	1.422
Antagonism	3.21 (2.64)	2.56 (2.25)	1.00	1.03	0.61	0.81	2.469	0.014 *	0.132	1.167
Detachment	5.89 (2.67)	4.80 (2.65)	0.17	0.54	−0.54	0.12	3.719	<0.001 **	0.515	1.67
Disinhibition	4.98 (2.80)	4.49 (2.35	0.59	0.30	0.57	−0.30	1.77	0.078	−0.054	1.032
Psychoticism	6.42 (3.13)	5.41 (2.97)	0.09	0.39	−0.58	−0.16	3.039	0.003 **	0.358	1.67
Brief-COPE										
Emotion-focused strategies	30.02 (5.22)	26.93 (5.17)	0.17	0.30	−0.22	−0.20	5.401	<0.001 **	1.966	4.217
Problem-focused strategies	20.98 (4.69)	20.36 (4.57)	0.28	0.08	−0.29	−0.31	1.232	0.219	−0.374	1.627
Avoidance coping	15.88 (3.20)	14.20 (3.07)	0.22	1.02	−0.18	2.20	4.879	<0.001 **	1.001	2.352

Note: Adverse Childhood Experiences Questionnaire (ACE-Q), General Anxiety Disorder-7 (GAD-7), Patient Health Questionnaire-9 (PHQ-9), Personality Inventory for DSM (PID-5), Confidence Interval (CI); * *p*-value < 0.05 and ** *p*-value < 0.01.

**Table 3 ijerph-20-05906-t003:** Bivariate Pearson’s correlation among the main variables of the study.

	1	2	3	4	5	6	7	8	9	10	11	12	13	14
ACE-Q	-													
2. Negative affectivity	0.118 *	-												
3. Detachment	0.227 **	0.287 **	-											
4. Antagonism	0.134 **	0.139 **	0.276 **	-										
5. Disinhibition	0.073	0.229 **	0.330 **	0.357 **	-									
6. Psychoticism	0.181 **	0.462 **	0.404 **	0.327 **	0.370 **	-								
7. Problem-focused coping	−0.004	0.062	−0.263 **	0.082	−0.027	0.081	-							
8. Emotion-focused coping	0.104 *	0.309 **	0.053	0.146 **	0.045	0.234 **	0.544 **	-						
9. Avoidance coping	0.280 **	0.427 **	0.332 **	0.282 **	0.272 **	0.467 **	0.143 **	0.374 **	-					
10. GAD-7	0.154 **	0.562 **	0.286 **	0.106 *	0.272 **	0.425 **	0.116 *	0.323 **	0.442 **	-				
11. PHQ-9	0.292 **	0.482 **	0.443 **	0.180 **	0.318 **	0.571 **	0.038	0.325 **	0.506 **	0.696 **	-			
12. Age	0.072	−0.129 *	0.015	−0.059	−0.076	−0.163 **	−0.056	−0.001	−0.082	−0.042	−0.071	-		
13. Gender	0.077	0.252 **	0.053	−0.116 *	0.023	0.079	−0.064	0.043	0.046	0.162 **	0.114 *	0.020	-	
14. Group	−0.187 **	−0.149 **	−0.189 **	−0.127 *	−0.091	−0.155 **	−0.064	−0.269 **	−0.245 **	−0.254 **	−0.404 **	−0.070	0.162 **	-

Note: Adverse Childhood Experiences Questionnaire (ACE-Q); General Anxiety Disorder-7 (GAD-7); Patient Health Questionnaire-9 (PHQ-9). Gender (1 = male, 2 = female). Group (1 = Counseling group, 2 = Non-counseling group). * *p* < 0.05 and ** *p* < 0.01.

**Table 4 ijerph-20-05906-t004:** Mediation and indirect effects with standardized estimates of coping strategies and personality traits in the relationship among ACE-Q, GAD-7, and PHQ-9.

Pathway	Estimate	SE	*z*	*p*	95% [CI]	Effect Size		
ACE-Q -> GAD-7						*P_M_*	SE	95%CI
Total effect	0.09	0.05	1.94	0.049 *	[0.00, 0.18]			
Direct effect	0.00	0.04	0.11	0.909	[−0.07, 0.08]			
Total indirect effect	0.09	0.03	2.64	<0.001 **	[0.02, 0.15]			
Specific indirect effect								
ACE-Q -> Avoidance -> GAD-7	0.03	0.01	2.35	0.019 *	[0.01, 0.07]	0.38	0.16	[0.07, 0.70]
ACE-Q -> PHQ-9								
Total effect	0.21	0.05	4.30	<0.001 **	[0.12, 0.31]			
Direct effect	0.10	0.04	2.37	0.02 *	[0.02, 0.18]			
Total indirect effect	0.12	0.03	3.67	<0.001 **	[0.05, 0.18]			
Specific indirect effects								
ACE-Q -> Detachment -> PHQ-9	0.03	0.01	2.55	0.011 *	[0.01, 0.06]	0.14	0.05	[0.03, 0.24]
ACE-Q -> Psychoticism -> PHQ-9	0.05	0.02	2.79	0.005 **	[0.02, 0.08]	0.22	0.08	[0.06, 0.38]
ACE-Q -> Avoidance ->PHQ-9	0.03	0.01	2.215	0.027 *	[0.01, 0.07]	0.15	0.07	[0.02, 0.29]

Note: SE = standard error; Adverse Childhood Experiences Questionnaire (ACE-Q); General Anxiety Disorder-7 (GAD-7), Patient Health Questionnaire-9 (PHQ-9), Ratio of the indirect effect to the total effect (P_m_); * *p*-value < 0.05 and ** *p*-value < 0.01.

## Data Availability

The data that support the findings of this study are available from the first or corresponding author upon request.

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
