# Peer review of "Adverse Childhood Experiences and Mental Health among Students Seeking Psychological Counseling Services"

_ijerph, 2023, doi:10.3390/ijerph20105906_

Round 1

Reviewer 1 Report

IJERPH-23129724- review

It was a pleasure reviewing the article “The Impact of Adverse Childhood Experiences on the Mental Health of University Students Seeking Psychological Counseling Services”. The authors discuss the significant issue of adverse childhood experiences and mental health. However, I do believe that some areas need refinement. The following are comments and suggestions section-wise.

Title:

The title needs modification to portray the objective of the study. The title shows that there is some special impact of ACE and mental health of university students who are receiving counseling services. However, the objective of the study is to highlight the significance of ACE screening to identify the risk of severe psychological disturbances. The first impression of the title seems to be that counseling services might be improving their status. In contrast, results show the worse psychological condition of UPC students. In fact, the UPC student group has just received an intake clinical interview that is not contributing to the mental health status. It can be better as, “Prevalence of Adverse Childhood Experiences on the Mental Health of University Students Seeking Psychological Counseling Services.”

Abstract:

1.     Line 11-17: Make a correction to ‘who had approached for counseling services’ in place of ….. who had a counseling session at UPC service.

2.     Line 17-18: Correct as ‘students who approached UPC services’ in the following sentence: We found that students who received counseling sessions scored….

3.     Line 18: This study is neither experimental nor quasi-experimental. Even a clinical intake interview cannot be considered a psychological intervention for the treatment purpose. Therefore, correct the control group as a non-counseling group. Use the terminology of counseling group and non-counseling group throughout the article write-up and tables.

Introduction:

1.     Line 38-40: Both of the studies (7-8) given in the citation do not fully prove the claim. These studies are discussing the higher association of trauma exposure among severe mental illness or psychopathology, not the higher occurrence compared to the general population. Mauritz et al (2013) studies are a systematic review of 33 studies with diagnosed patients to highlight the significance of adverse childhood experience recognition among psychiatric patients. However, need to be replaced with some of the latest studies as it is reviewing 1980-2010 studies and is based upon DSM-IV criteria. The study of Kessler et al.  (2010) also needs to be replaced with the latest studies due to DSM-IV.

2.     Line 62-64: Add citation for the following claim. ‘Thus, the services offered by the university can help students develop healthy relationships among peers, improve self-esteem, overcome personal obstacles, build resilience, reduce psychological distress, and prevent more severe issues from arising.’

3.     Line 84-87: convert the citation according to the journal style.

Material and methods:

1.     Line 125: Complete the following sentence. ‘The participants were invited to participate in the online survey by filling out the Google.’

2.     Line 138: Make sure to correct the counseling group in place of the experimental group.

3.     Line 143-144: Complete the sentence by adding: ‘It was conveyed to the participants that accepting or…….

4.     Line 147-149: Exclusion criteria are not clear (As exclusion criteria, we did not include students with a  significant mental or physical condition affecting their ability to complete the questionnaire accurately). The study is measuring GAD, depression, and personality disorders.  Clarification is needed.

5.     Add the number of participants in the counseling group and non-counseling group in the participants’ section.

6.     Add sampling techniques, strategy, and sample size formula.

7.      Provide the exactly reported reliability coefficient by other studies for the scales that are missing for ACE-Q, GAD-7, PHQ-9, and Brief-COPE.

8.     Line 170-171: There is a need to mention the cutoff score for ACE-Q as there are two groups for comparison. Scores of 4 or more are considered clinically significant. ‘A minority (5%–10%) of the general population score 4 or more, where the general long-term health consequences become most pronounced (Hughes et al., 2017).’

9.     Line 175-176: How 0.73 is demonstrating good reliability?

10.  Line 200-210: Recheck the details for PID-5. The full scale has 220 items and the brief scale has 25 items. It has a 4-point Likert scale. Response categories are Very False or Often False to Very True or Often True. Correct the calculated reliability for PID-5 (Line 209-210: We found that the GAD-7 has good reliability (Cronbach’s α is 0.83). Update reference 32 for PID-5: Krueger R. F., Derringer J., Markon K. E., Watson D., Skodol A. E. (2013). The Personality Inventory DSM-5-Brief Form (PID-5-BF) -Adult. American Psychiatric Association.

Data Analysis:

Line 250-251: Convert the citation according to the journal style [35] and update the reference year as 2016.

Results:

1.     Line 261-263: There is a discrepancy in the description and table: The mean age of the counseling group was 21.96 ± 2.9 years, ranging from 18 to 35 years; while the mean age of the control group was 21.41 ± 3.9 years, ranging from 18 to 50 years (see Table 2). No significant difference was found between groups regarding age (p = 0.385).

2.     Table 2 and figure 1 analysis needs to be performed again. The comparison should be restricted only to the counseling group and those who meet the criteria of ACE-Q (score 4 or above) from the non-counseling group.  

3.     Line 358: Correct Figure 2 in place of 1.

 References:

Incomplete references: 5, 7, 9, 16, 22, 24, 27, 28, 31, 35 (2016), 36, 44, 45, 46, 48, 51, 52, 54, 57

Author Response

Reviewer #1:

It was a pleasure reviewing the article “The Impact of Adverse Childhood Experiences on the Mental Health of University Students Seeking Psychological Counseling Services”. The authors discuss the significant issue of adverse childhood experiences and mental health. However, I do believe that some areas need refinement. The following are comments and suggestions section-wise.

Thank you for your helpful critique of our manuscript.

-Title: The title needs modification to portray the objective of the study. The title shows that there is some special impact of ACE and mental health of university students who are receiving counseling services. However, the objective of the study is to highlight the significance of ACE screening to identify the risk of severe psychological disturbances. The first impression of the title seems to be that counseling services might be improving their status. In contrast, results show the worse psychological condition of UPC students. In fact, the UPC student group has just received an intake clinical interview that is not contributing to the mental health status. It can be better as, “Prevalence of Adverse Childhood Experiences on the Mental Health of University Students Seeking Psychological Counseling Services.”

The Reviewer is correct, and we have edited the title of the manuscript.

 Abstract:

  1. Line 11-17: Make a correction to ‘who had approached for counseling services’ in place of ….. who had a counseling session at UPC service.

We have edited it as requested.

  1. Line 17-18: Correct as ‘students who approached UPC services’ in the following sentence: We found that students who received counseling sessions scored….

We have edited it as requested.

  1. Line 18: This study is neither experimental nor quasi-experimental. Even a clinical intake interview cannot be considered a psychological intervention for the treatment purpose. Therefore, correct the control group as a non-counseling group. Use the terminology of counseling group and non-counseling group throughout the article write-up and tables.

We have edited it as requested.

Introduction:

  1. Line 38-40: Both of the studies (7-8) given in the citation do not fully prove the claim. These studies are discussing the higher association of trauma exposure among severe mental illness or psychopathology, not the higher occurrence compared to the general population. Mauritz et al (2013) studies are a systematic review of 33 studies with diagnosed patients to highlight the significance of adverse childhood experience recognition among psychiatric patients. However, need to be replaced with some of the latest studies as it is reviewing 1980-2010 studies and is based upon DSM-IV criteria. The study of Kessler et al.  (2010) also needs to be replaced with the latest studies due to DSM-IV.

We thank the Reviewer#1 for encouraging us to consider new literature. We have edited the sentence and added new recent references.

  1. Line 62-64: Add citation for the following claim. ‘Thus, the services offered by the university can help students develop healthy relationships among peers, improve self-esteem, overcome personal obstacles, build resilience, reduce psychological distress, and prevent more severe issues from arising.’

We have added new recent references.

  1. Line 84-87: convert the citation according to the journal style.

We have edited it as requested.

Material and methods:

  1. Line 125: Complete the following sentence. ‘The participants were invited to participate in the online survey by filling out the Google.’

We have edited it as requested.

  1. Line 138: Make sure to correct the counseling group in place of the experimental group.

We thank Reviewer#1 for pointing out this issue. We have modified It as requested.

  1. Line 143-144: Complete the sentence by adding: ‘It was conveyed to the participants that accepting or…….

We have edited it as requested.

  1. Line 147-149: Exclusion criteria are not clear (As exclusion criteria, we did not include students with a  significant mental or physical condition affecting their ability to complete the questionnaire accurately). The study is measuring GAD, depression, and personality disorders.  Clarification is needed.

The Reviewer is correct. We have added new details in the method section.

  1. Add the number of participants in the counseling group and non-counseling group in the participants’ section.

We have edited it as requested.

  1. Add sampling techniques, strategy, and sample size formula.

We thank Reviewer#1 for pointing out this issue. In the method session we added the required information on technique and strategy. Regarding the sample size formula, we added the calculation and the criteria used. To define the sample size, we performed an a priori power analysis. However, it was not included in the manuscript because we considered this a preliminary step. Following your suggestion, we included a paragraph describing the analysis in the statistical section.

  1. Provide the exactly reported reliability coefficient by other studies for the scales that are missing for ACE-Q, GAD-7, PHQ-9, and Brief-COPE.

As reported by several studies, we report the total values which indicate the reliability of the scales used.

  1. Line 170-171: There is a need to mention the cutoff score for ACE-Q as there are two groups for comparison. Scores of 4 or more are considered clinically significant. ‘A minority (5%–10%) of the general population score 4 or more, where the general long-term health consequences become most pronounced (Hughes et al., 2017).’

We have edited it as requested.

  1. Line 175-176: How 0.73 is demonstrating good reliability?

We have edited the sentence by adding details and reference.

  1. Line 200-210: Recheck the details for PID-5. The full scale has 220 items and the brief scale has 25 items. It has a 4-point Likert scale. Response categories are Very False or Often False to Very True or Often True. Correct the calculated reliability for PID-5 (Line 209-210: We found that the GAD-7 has good reliability (Cronbach’s α is 0.83). Update reference 32 for PID-5: Krueger R. F., Derringer J., Markon K. E., Watson D., Skodol A. E. (2013). The Personality Inventory DSM-5-Brief Form (PID-5-BF) -Adult. American Psychiatric Association.

We thank the Reviewer for noticing this typo. We have edited it. 

 Data Analysis:

Line 250-251: Convert the citation according to the journal style [35] and update the reference year as 2016.

We have edited it as requested.

 Results:

  1. Line 261-263: There is a discrepancy in the description and table: The mean age of the counseling group was 21.96 ± 2.9 years, ranging from 18 to 35 years; while the mean age of the control group was 21.41 ± 3.9 years, ranging from 18 to 50 years (see Table 2). No significant difference was found between groups regarding age (p = 0.385).

We have deleted Table 2. It was a typo.

  1. Table 2 and figure 1 analysis needs to be performed again. The comparison should be restricted only to the counseling group and those who meet the criteria of ACE-Q (score 4 or above) from the non-counseling group.  

The Reviewer#1 raises a very critical issue. In the Result section we now state that “The 35% of students in the counseling group scored 4 or above on the ACEs-Q questionnaire, compared to the 20% in the control group”. The aim of the study was to make an initial analysis of the phenomenon among the students on the Campus. Using the cutoff of 4 or higher could probably be of great interest to clinical and larger populations.

  1. Line 358: Correct Figure 2 in place of 1.

We have edited it as requested.

 References:

Incomplete references: 5, 7, 9, 16, 22, 24, 27, 28, 31, 35 (2016), 36, 44, 45, 46, 48, 51, 52, 54, 57

We have updated the bibliography using the journal guidelines via Mendeley - Reference Management Software

My coauthors and I thank you for considering the revised manuscript. 

We look forward to hearing from you.

Sincerely yours,

FC

Reviewer 2 Report

Dear Authors,

Thanh you very much for your research, I have some changes to propose to you.

TITLE – please add: University of Calabria experience

Line 58 and 372 - change ‘psychological distress’ in ‘mental health’ – the title of your paper wants to examinate the role of ACE exposure on the mental health. According to your hypothesis, the psychological distress is one of the dimensions of the mental health, along with personality traits and coping strategies. In your paper the psychological distress is described in two dimensions – anxiety and depressive symptoms. Otherwise consider changing the title.

Line 62-64 – reconsider the phrase – no data to support the phrase

Line 70 – add: on students’ mental health

Line 100 – change the phrase – needs data. We don’t know that. Consider resilience

Line 109-111- please consider changing 'well-being' in mental health

paragraph Measures - how many questions where there all together? what was the time to answer all the questions in the survey? add some more description. 

Line 171 – 173 – phrase:’ The ACE-Q is widely used in public health and medical research, as well as in clinical settings, to help identify and address the impact of childhood trauma on a person’s health and well-being’ – consider moving to the line 162 

Lines 200 – 210 – please review the whole paragraph. there are inconsistencies in the description of the tool. The last sentence of the paragraph describes different tool. Please revise the name of the tool you describe. –for example in the abstract, in line 160, 238, 456, page 8 you use PID-5. In the paragraph lines 200 – 210 you use PID-5-BF. Please make sure that you use correct name of the measuring tool. PID-5 and PID-5-BF are different tools. 

Line 398, 424 – add reference number instead of the year

Line 461 – 463 – please reconsider the phrase. What it means ‘ideal setting’? UPC is important for screening and identifying students at higher risk and should give equal access to all, not ideal.

Line 476- 478 – I do not agree that seeking the way to overcome the barriers that inhibit students from seeking UPC services is crucial. As you underline in line 437, some coping mechanism are a protecting factors even if considered maladaptive. It is important to identify and address the external barriers (organizational, institutional) to let the UPC services available for all. Furthermore, what do you mean by ‘optimal psychological outcomes’I don’t agree that support through UPC services aims to optimal academic achievement or optimal psychological outcomes? please reconsider the phrase. It may lead to hidden bias.

Line 485 – consider changing ‘mental health treatment’ in ‘mental health support services’

Limitations- please reconsider the paragraph. Might be important to add the resilience mechanisms theory. 

Author Response

TITLE – please add: University of Calabria experience

We would like to thank the Reviewer#2 for this comment. Now we have changed the title following the suggestions of another reviewer in “Prevalence of Adverse Childhood Experiences on the Mental Health of University Students Seeking Psychological Counseling Services”. In the method we have specified that the participants are students of the University of Calabria.

-Line 58 and 372 - change ‘psychological distress’ in ‘mental health’ – the title of your paper wants to examinate the role of ACE exposure on the mental health. According to your hypothesis, the psychological distress is one of the dimensions of the mental health, along with personality traits and coping strategies. In your paper the psychological distress is described in two dimensions – anxiety and depressive symptoms. Otherwise consider changing the title.

Now we have changed the title in “Prevalence of Adverse Childhood Experiences on the Mental Health of University Students Seeking Psychological Counseling Services”.

-Line 62-64 – reconsider the phrase – no data to support the phrase

We have edited it as requested.

-Line 70 – add: on students’ mental health

We have edited it as requested.

-Line 100 – change the phrase – needs data. We don’t know that. Consider resilience

We have added new recent references. Furthermore, we have considered resilience in the limits section.

-Line 109-111- please consider changing 'well-being' in mental health

We have edited it as requested.

-paragraph Measures - how many questions where there all together? what was the time to answer all the questions in the survey? add some more description. 

We would like to thank the Reviewer#2 for this comment. New details have been added in the method as "The online survey with 118 questions" or "Estimated time to complete the questionnaire was approximately 15-20 minutes."

-Line 171 – 173 – phrase:’ The ACE-Q is widely used in public health and medical research, as well as in clinical settings, to help identify and address the impact of childhood trauma on a person’s health and well-being’ – consider moving to the line 162 

We have edited it as requested.

-Lines 200 – 210 – please review the whole paragraph. there are inconsistencies in the description of the tool. The last sentence of the paragraph describes different tool. Please revise the name of the tool you describe. –for example in the abstract, in line 160, 238, 456, page 8 you use PID-5. In the paragraph lines 200 – 210 you use PID-5-BF. Please make sure that you use correct name of the measuring tool. PID-5 and PID-5-BF are different tools. 

We thank the Reviewer for noticing this typo. We have edited it as requested.

-Line 398, 424 – add reference number instead of the year

We have edited it as requested.

-Line 461 – 463 – please reconsider the phrase. What it means ‘ideal setting’? UPC is important for screening and identifying students at higher risk and should give equal access to all, not ideal.

We thank Reviewer#2 for pointing out this issue. Now, the sentence was edited.

-Line 476- 478 – I do not agree that seeking the way to overcome the barriers that inhibit students from seeking UPC services is crucial. As you underline in line 437, some coping mechanism are a protecting factors even if considered maladaptive. It is important to identify and address the external barriers (organizational, institutional) to let the UPC services available for all. Furthermore, what do you mean by ‘optimal psychological outcomes’I don’t agree that support through UPC services aims to optimal academic achievement or optimal psychological outcomes? please reconsider the phrase. It may lead to hidden bias.

We thank you for pointing out this issue. We have edited the sentence following Reviewer#2's suggestion.

-Line 485 – consider changing ‘mental health treatment’ in ‘mental health support services’.

We have edited it as requested.

-Limitations- please reconsider the paragraph. Might be important to add the resilience mechanisms theory. 

We have edited it as requested. We have considered resilience in the limits section.

My co-authors and I thank you for considering the revised manuscript. 

We look forward to hearing from you.

Sincerely yours,

FC

Reviewer 3 Report

First I would like to thank Editor of the Journal (Int. J. Environ. Res. Public Health) for giving me a chance to review the research article entitled The Impact of Adverse Childhood Experiences on the Mental  Health of University Students Seeking Psychological Counseling Services

Point 1: I consider this research article is relevant, well-planned and well-developed research. It is simple and clear. I  miss a clearer and more detailed description of the sampling process.

Point 2. Some references are missing in the reference section, see page 15 line 522

Point 3. Maintain the reference style as you wrote some reference in APA syles

Point 4. Tables should be modified in order to avoid overlapping

Finally, I would like to congratulate the authors, I know it is difficult to publish such type of studies but I consider the quality of the data and their veracity to be more important criteria.

Author Response

First I would like to thank Editor of the Journal (Int. J. Environ. Res. Public Health) for giving me a chance to review the research article entitled The Impact of Adverse Childhood Experiences on the Mental  Health of University Students Seeking Psychological Counseling Services

Thank you for your helpful critique of our manuscript.

-Point 1: I consider this research article is relevant, well-planned and well-developed research. It is simple and clear. I  miss a clearer and more detailed description of the sampling process.

 The Reviewer is correct. We have added new details in the method section.

-Point 2. Some references are missing in the reference section, see page 15 line 522

We have edited it as requested.

-Point 3. Maintain the reference style as you wrote some reference in APA syles

We have edited it as requested.

-Point 4. Tables should be modified in order to avoid overlapping

 Some errors have been fixed

My coauthors and I thank you for considering the revised manuscript. 

We look forward to hearing from you.

Sincerely yours,

FC